Methods

# Rapid and precise genotyping of transgene zygosity in mice using an allele-specific method

Jianqi Yang[1], Alison N DeVore[1], Daniel A Fu[1], Mackenzie M Spicer[1], Mengcheng Guo[1], Samantha G Thompson[1], Katelin E Ahlers-Dannen[1], Federica Polato[3], Andre Nussenzweig[3], Rory A Fisher[1,2]

Precise determination of transgene zygosity is essential for use of transgenic mice in research. Because integration loci of transgenes are usually unknown due to their random insertion, assessment of transgene zygosity remains a challenge. Current zygosity genotyping methods (progeny testing, qPCR, and NGS-computational biology analysis) are time consuming, prone to error or technically challenging. Here, we developed a novel method to determine transgene zygosity requiring no knowledge of transgene insertion loci. This method applies allele-specific restriction enzyme digestion of PCR products (RE/PCR) to rapidly and reliably quantify transgene zygosity. We demonstrate the applicability of this method to three transgenic strains of mice (*Atm* Tg$^{C3001L}$, *Nes-Cre*, and *Syn1-Cre*) harboring a unique restriction enzyme site on either the transgene or its homologous sequence in the mouse genome. This method is as accurate as the gold standard of progeny testing but requires 2 d instead of a month or more. It is also exceedingly more accurate than the most commonly used approach of qPCR quantification. Our novel method represents a significant technical advance in determining transgene zygosities in mice.

## Introduction

Since the pioneering work creating transgenic mice ~40 years ago (Gordon et al, 1980; Brinster et al, 1981; Costantini & Lacy 1981), transgenic mouse models continue to be a powerful and indispensable tool in virtually all fields of biological research. Transgenic mice are created by the random insertion of foreign DNA into the mouse genome via either microinjection or retroviral infection methods (Gordon et al, 1980; Brinster et al, 1981; Costantini & Lacy 1981; Lois et al, 2002). Precise assessment of transgene zygosity in the mouse genome is highly desirable for cost-effective management of mouse colonies and as a time-saving strategy in breeding mice for preparation of study cohorts. In addition, the ability to determine whether mice are heterozygous or homozygous for a transgene enables transgene dose effect studies and avoids potential insertional mutagenesis effects in homozygous mice. Hence, rapid and reliable techniques to detect the zygosity of transgenic mice are highly sought research tools.

Assessment of transgene zygosity poses a technical challenge because the flanking sequence of the insertion loci is unknown when foreign DNA is randomly inserted into the mouse genome. Standard PCR (Polymerase Chain Reaction) protocols that detect gene zygosity by selecting one PCR primer on the flanking sequence of an inserted DNA are not applicable to transgenes, unless the flanking sequence of the transgene is deciphered by special means, such as via whole genome sequencing (Yong et al, 2015) or transgene insertion site mapping (Irie et al, 2017). To date, progeny testing (McHugh et al, 2012), quantitative Southern blotting analysis (Stumpel et al, 2018), fluorescent in situ hybridization (McHugh et al, 2012), direct fluorescence imaging (Lin et al, 2015), and quantitative PCR (qPCR) assay (Tesson et al, 2010) are commonly used methods to discriminate homozygous from heterozygous transgenic mice. Computation biology-based methods such as Illumina whole genome sequencing (Yong et al, 2015; Irie et al, 2017), nanopore sequencing (Giraldo et al, 2020), Xdrop indirect sequencing (Samplix) (Blondal et al, 2021), and nanopore adaptive sampling (Ulrich et al, 2022) have recently demonstrated their abilities in determination of transgene zygosity. However, these methods are either time-consuming, or require knowledge and capability of biological computation. For instance, progeny testing (McHugh et al, 2012)—the gold standard of precise detection of zygosity of any gene—is based on Mendel's law of segregation so that all offspring of a crossing between homozygous and WT mice are heterozygous. As a result, like quantitative Southern blotting analysis and fluorescent in situ hybridization, progeny testing is a labor-intensive and time-consuming assay that is not suitable for large-scale applications. Direct fluorescence imaging is limited to mouse models that harbor a fluorescence reporter transgene (Lin et al, 2015), and is thereby not applicable to most of existing nonfluorescent transgenic mouse models. Computation biology-based methods require in-depth knowledge and skill with bioinformatics. Lastly, qPCR assay has been successfully used to determine transgene zygosity in various species (Mason et al, 2002; Wang et al, 2015), including mice (Tesson et al, 2010). Yet, studies have shown that qPCR assays tend to generate ambiguous results (Bubner et al, 2004). A fast and reproducible

[1]Departments of Neuroscience and Pharmacology, The University of Iowa, Iowa City, IA, USA   [2]Roy J and Lucille A Carver College of Medicine, The University of Iowa, Iowa City, IA, USA   [3]Laboratory of Genome Integrity, National Institutes of Health, Centre for Cancer Research, Bethesda, MD, USA

Correspondence: rory-fisher@uiowa.edu

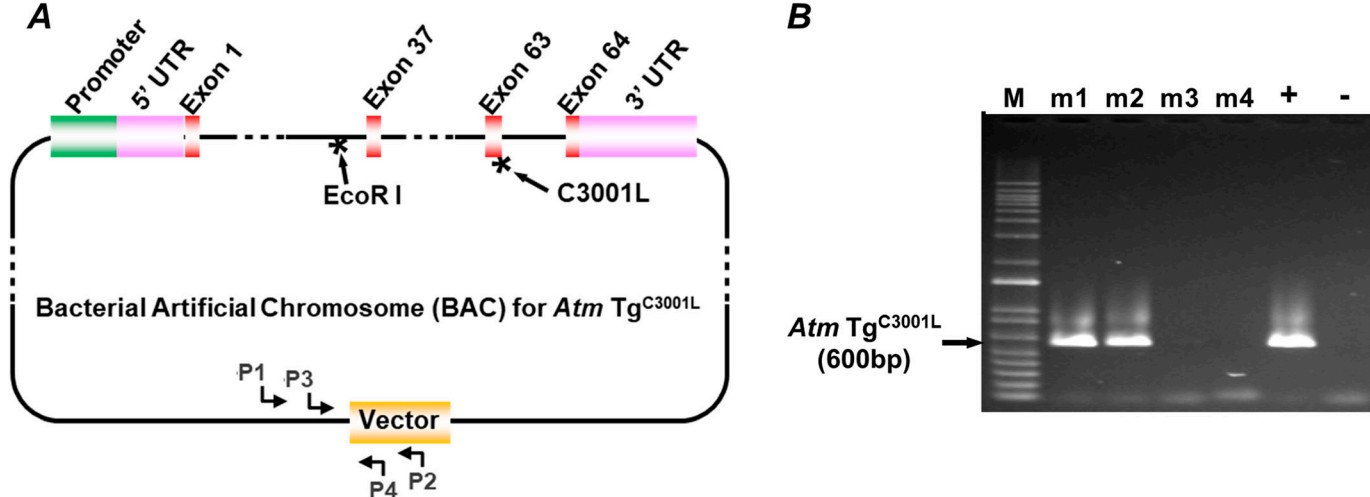

**Figure 1. Creation of *Atm* Tg^C3001L transgenic mice.**
**(A)** A schematic illustration of *Atm* transgene in a bacterial artificial chromosome. The *Atm* Tg^C3001L transgene was constructed using a recombineering technique in a bacterial artificial chromosome. Compared with the mouse WT ATM gene, *Atm* Tg^C3001L transgene contains two mutations (highlighted by *), one encoding C3001L located in exon 63 and the other an intronic EcoRI site located 185 bp upstream of exon 37. Primers P1 and P2 are used to determine the presence of *Atm* Tg^C3001L transgene, and primers P3 and P4 are for qPCR-based assay to genotype the zygosity of *Atm* Tg^C3001L transgene. **(B)** A representative PCR genotyping result of *Atm* Tg^C3001L transgenic mice using primers P1 and P2. +, positive control; –, negative control; M, 1 Kb plus DNA ladder (Invitrogen); m1–m4, mouse 1–4, are offspring from crossing a *Atm* Tg^C3001L +/– mouse to a WT mouse. M1 and m2 are positive for the transgene and m3 and m4 are negative.

method for the determination of transgene zygosity is still greatly needed by the scientific community.

The serine threonine kinase ataxia telangiectasia mutated (ATM) is a central regulator of double-strand DNA break signaling (Lee & Paull 2021). Mutations leading to loss of ATM in humans cause ataxia telangiectasia, a disease characterized by cerebellar degeneration, telangiectasia, immunodeficiency, cancer susceptibility, and radiation sensitivity (Amirifar et al, 2019). ATM is activated both by DNA double-strand breaks and by oxidation of a critical Cys (3001 in mouse) (Guo et al, 2010; Lee & Paull 2021). To study the role of oxidative activation of ATM, Federica Polato and Andre Nussenzweig created an oxidation-resistant ATM mutant mouse strain (*Atm* Tg^C3001L) by recombineering (Yu et al, 2000; Gong & Yang 2005) *Atm* gene in a bacterial artificial chromosome (BAC) clone. When using qPCR assay (Tesson et al, 2010) to determine the transgene zygosity in these mice, we encountered considerable in-consistency and mis-genotyping. To resolve this issue, we developed a new allele-specific quantification approach to determine the zygosity of *Atm* Tg^C3001L mice that is simple, fast, and reliable. We further demonstrated that this method is applicable to other transgenic models, such as *Nes-Cre* mice (Strain 016261; Jackson laboratory) and *Syn1-Cre* mice (Strain 003966; Jackson laboratory). The requirements for this method to work are (1) a transgene containing homology sequence to its mouse counterpart; (2) a unique restriction enzyme site on either the transgene or its homologous mouse sequence.

## Results

### Generation of *Atm* Tg^C3001L transgenic mouse line

The discovery that ATM is activated by oxidative stress independent of DNA double-strand breaks (Guo et al, 2010) prompted the development of an oxidation-resistant ATM transgenic mouse line by Federica Polato and Andre Nussenzweig. Guo et al first showed that Cys 2991 of human ATM (Cys 3001 in mouse ATM) mediated ATM activation in vitro in response to oxidative stress (Guo et al, 2010). To study the role of oxidative activation of ATM in vivo, a transgene was constructed using a BAC vector. Two mutations were introduced into the transgene construct using a recombineering technique (Yu et al, 2000; Gong & Yang 2005): a Cys3001 to Leu within exon 63 of the mouse ATM gene, and an intronic EcoRI restriction enzyme site located at 185 bp upstream of exon 37 (Fig 1A). After microinjection of *Atm* Tg^C3001L DNA into mouse oocytes, a positive founder was identified, backcrossed on to C57BL/6 background, and its offspring (m1 and m2) showed stable transmission of the *Atm* Tg^C3001L transgene (Fig 1B).

### Inaccuracy in qPCR-based *Atm* Tg^C3001L zygosity determination

To determine *Atm* Tg^C3001L zygosity in mice, we first followed a commonly used qPCR-based method by which transgene zygosity is determined according to their qPCR $2^{-\Delta\Delta Ct}$ values (Tesson et al, 2010). Housekeeping gene 18S ribosomal RNA (18S rRNA) was used as the reference control. The qPCR specificity for *Atm* Tg^C3001L transgene was achieved by selecting primer 4 from the vector sequence of BAC, which shares a low similarity to the mouse genomic sequence (Fig 1A). Given short amplicons are typically amplified with high efficiency (Bustin & Huggett 2017), short 54- and 77-bp amplicons (116 and 146-bp qPCR products, respectively) were selected for *Atm* Tg^C3001L transgene and reference gene 18S rRNA, respectively. Indeed, Fig 2 shows both qPCRs were robust and specific. When this qPCR protocol was used to determine *Atm* Tg^C3001L zygosity in a cohort of 105 mice, considerable error and ambiguity were found.

As an example, Table 1 shows raw and calculated data of qPCRs for one litter of seven mice from a breeding of heterozygous parents. The qPCR was performed in triplicate for each DNA sample. The last column of this table shows *Atm* Tg$^{C3001L}$ genotype determined using the qPCR method. Although this qPCR method correctly genotyped all control mice (controls 1–4), mice m5, m6, and m10 were of an unknown genotype due to their $2^{-\Delta\Delta Ct}$ values being out of the recommended $2^{-\Delta\Delta Ct}$ cut-off ranges of 0.8–1.3 and 1.8–2.3 for heterozygous and homozygous mice (Tesson et al, 2010), respectively (Table 1). Furthermore, mouse m9 was mis-genotyped as homozygous by the qPCR-based method, because a following-up progeny testing result demonstrated that m9 is a heterozygous mouse (Fig 3). Table 2 summarizes the assignments of *Atm* Tg$^{C3001L}$ genotypes to the 105 mice assayed with the qPCR protocol. 48 of 105 mice were of an unknown genotype because their $2^{-\Delta\Delta Ct}$ values were out of the recommended $2^{-\Delta\Delta Ct}$ cut-off ranges (Tesson et al, 2010). In addition, 9 of 19 qPCR-assigned homozygous mice proved to be heterozygous mice when followed up with progeny testing. In summary, our results agree with the findings from other reports that qPCR-based zygosity genotyping is often ambiguous and inaccurate (Bubner et al, 2004; Ji et al, 2005; Mieog et al, 2013).

### Rapid determination of transgene zygosity using an allele-specific genotyping method

We sought to develop a novel and reproducible alternative method to genotype *Atm* Tg$^{C3001L}$ zygosity by taking advantage of the EcoRI site that was introduced into the *Atm* Tg$^{C3001L}$ transgene. First, a PCR was designed to simultaneously amplify the EcoRI-containing segment of *Atm* Tg$^{C3001L}$ transgene and its homologous endogenous WT *Atm* gene (Fig 4A). The amplification efficiency for both templates is presumably the same in a single PCR reaction, as these two templates are identical in sequence except for two nucleotides at the EcoRI site. Therefore, the template ratio, which is the transgene versus endogenous gene ratio (Tg/WT ratio), is equal to the yield ratio of the corresponding PCR products. Given that EcoRI can only cut the PCR product from *Atm* Tg$^{C3001L}$ transgene into 354 and 216-bp fragments, the PCR products generated from *Atm* Tg$^{C3001L}$ and endogenous WT *Atm* gene can be separated by agarose gel electrophoresis (Fig 4B). By quantification of these DNA bands, the yield ratio, or the Tg/WT template ratio, can be readily calculated. Second, the transgene is often inserted into the genome in a tandem repeat fashion, whereas its endogenous homolog is present in either one (+/−) or two copies (+/+) in the same genome. To prevent a situation where the majority of the PCR products are generated from the transgene due to its high tandem repeat, it is necessary to assess the repeat number of the transgene as compared with its endogenous homolog. An allele-specific RE/PCR was performed using *Atm* Tg$^{C3001L \ +/−}$ DNA that had been sequentially diluted with DNA extracted from C57BL/6 mice (*Atm*$^{+/+}$) (Fig S1A). Results showed that less than four repeats of *Atm* Tg$^{C3001L}$ transgene are present in the *Atm* Tg$^{C3001L}$ mouse genome, as an equal mixture of *Atm* Tg$^{C3001L \ +/−}$ DNA and C56BL/6 DNA (1 to 2 dilution) resulted in a marked decrease in the PCR product generated from the *Atm* Tg$^{C3001L}$ transgene (Fig S1A, lane 2). Indeed, Western blotting showed comparable expression levels of WT and C3001L mutants of ATM in mouse thymus (data not shown).

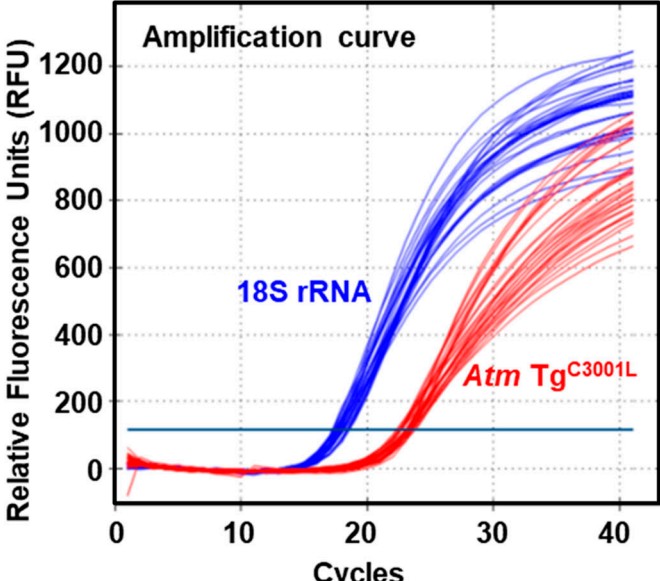

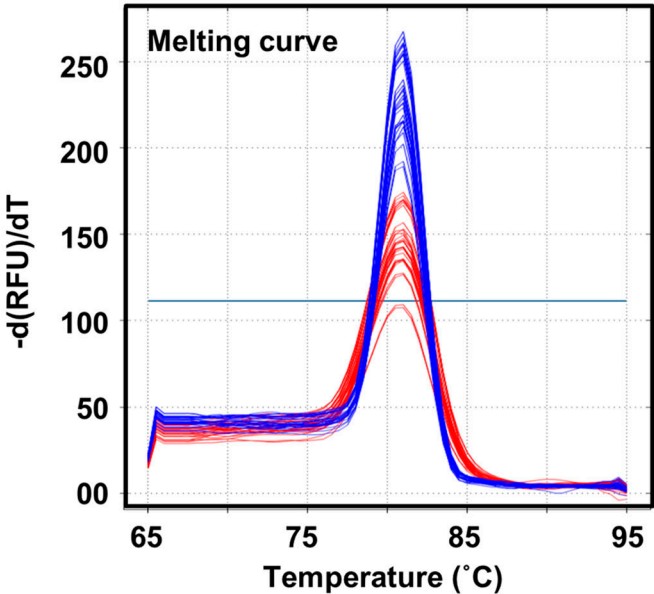

**Figure 2. qPCRs are robust and specific to both Atm Tg$^{C3001L}$ transgene and reference control of 18S rRNA.**
Representative qPCR amplification and melting curves for *Atm* Tg$^{C3001L}$ transgene (red) and 18S rRNA control (blue).

Furthermore, no dilution of DNA is needed, as both *Atm* Tg$^{C3001L}$ transgene and endogenous WT *Atm* are efficiently amplified in lane 1 (Fig S1A).

As a proof of concept, Fig 4B shows how allele-specific RE/PCR determines transgene homozygosity. Control mouse C5, which does not have an endogenous WT *Atm* gene, is used to show a complete EcoRI digestion of the PCR product from the *Atm* Tg$^{C3001L}$ transgene. The remaining mice (C6, C7, and m12–m15) have two copies of the endogenous WT *Atm* gene for a simplified analysis. The control mouse C7 was used to establish the Tg/WT ratio for heterozygous

**Table 1. Determination of *Atm* Tg^C3001L transgene zygosity using the qPCR-based method.**

| Mouse ID | Ct of *Atm* Tg^C3001L | | | Ct of 18S rRNA | | | ΔCt | ΔΔCt | 2^−ΔΔCt a | ATM Tg genotype |
|---|---|---|---|---|---|---|---|---|---|---|
| M5 | 22.2 | 22.2 | 22.2 | 17.0 | 17.0 | 17.1 | 5.17 | −0.78 | 1.72 | Unable to determine |
| M6 | 22.1 | 21.7 | 21.7 | 17.6 | 16.7 | 17.4 | 4.60 | −1.35 | 2.55 | Unable to determine |
| M7 | 23.0 | 23.0 | 22.9 | 18.3 | 18.2 | 18.1 | 4.77 | −1.18 | 2.27 | +/+ |
| M8 | 22.7 | 22.8 | 22.6 | 17.2 | 17.0 | 17.1 | 5.60 | −0.35 | 1.27 | +/− |
| M9 | 22.3 | 22.6 | 22.6 | 17.4 | 17.4 | 17.6 | 5.03 | −0.92 | 1.89 | +/+ |
| M10 | 22.7 | 23.0 | 22.6 | 17.4 | 17.2 | 17.2 | 5.50 | −0.45 | 1.37 | Unable to determine |
| M11 | 23.1 | 23.4 | 23.3 | 17.7 | 17.7 | 17.7 | 5.57 | −0.38 | 1.30 | +/− |
| Control 1 | 24.5 | 24.5 | 24.4 | 19.0 | 18.6 | 18.7 | 5.70 | −0.25 | 1.19 | +/− |
| Control 2 | 23.4 | 23.7 | 23.5 | 17.1 | 17.6 | 17.3 | 6.20 | 0.25 | 0.84 | +/− |
| Control 3 | 22.0 | 21.9 | 23.1 | 17.6 | 17.4 | 17.4 | 4.87 | −1.08 | 2.12 | +/+ |
| Control 4 | 37.0 | 39.6 | 35.1 | 16.5 | 16.6 | 16.5 | 20.70 | 14.75 | 0.00 | −/− |

[a]The cut-off thresholds of $2^{-\Delta\Delta Ct}$ for heterozygous and homozygous mice are 0.8–1.3 and 1.8–2.3, respectively (Tesson et al, 2010).

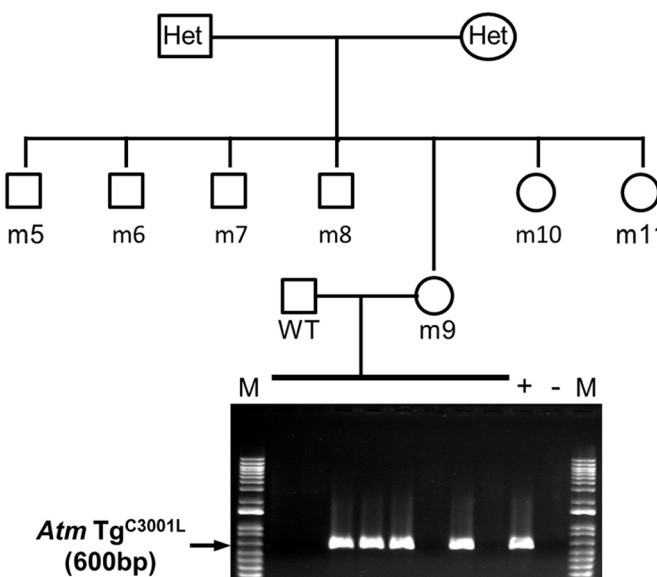

**Figure 3. Progeny testing demonstrated a failure in qPCR-based zygosity determination.**
A crossing of *Atm* Tg^C3001L heterozygous parents produced seven mice, m5 to m11. Progeny testing and genotyping results showed that mouse m9 was heterozygous, not homozygous, as incorrectly determined by the qPCR method (Table 1). This is because only four out of eight mice resulting from crossing m9 with a WT mouse were positive for the transgene rather than the expected 8 out of 8 if m9 was homozygous for the transgene. M, 1 Kb plus DNA ladder (Invitrogen); +, *Atm* Tg^C3001L+/− positive control mouse; −, *Atm* Tg^C30001L−/− negative control mouse.

**Table 2. Errors in qPCR-based determination of Atm Tg zygosity.**

| qPCR-based assignments of *Atm* Tg zygosity | # of mice | # of mice incorrectly genotyped by the qPCR method |
|---|---|---|
| −/− | 11 | 0 |
| +/− | 27 | 0 |
| +/+ | 19 | 9 |
| Unable to determine | 48 | N/A |

specific genotyping method (Fig S2A). Using this method, Fig 4C shows a representative result of genotyping *Atm* Tg^C3001L transgene in mice m16–m22, whereas C8–C12 are controls. Out of 105 mice genotyped using the qPCR method, 52 mice were re-genotyped using our allele-specific RE/PCR method. Of these 52 mice, only two homozygous mice were mis-genotyped as heterozygous due to a partial digestion of the PCR product by EcoRI. This result demonstrates an overall accuracy of the allele-specific genotyping method of more than 96%.

We next demonstrated the applicability of this method of zygosity determination to other homozygous viable transgenic mouse strains: *Nes-Cre* mice (Strain 016261; Jackson laboratory) and *Syn1-Cre* mice (Strain 003966; Jackson laboratory). These transgenic strains were created using rat DNAs, rat nestin genomic fragment (Zimmerman et al, 1994; Beech et al, 2004; Lagace et al, 2007) and rat synapin 1 gene promoter sequence (Sauerwald et al, 1990; Zhu et al, 2001), both of which are highly similar to the homologous mouse nestin and synapsin 1 sequences, respectively (Figs 5A and 6A). This enabled us to identify restriction enzyme sites unique to either the transgene or endogenous gene and PCR primers that recognize both rat and mouse DNA templates. For *Nes-Cre* mice, we selected primers 7 and 8, targeting both mouse and rat nestin intron 2 sequences (Fig 5A). A BglII site that is only present in the mouse template was used to cut the PCR

transgenic mice. Mice m12 to m15 are of unknown *Atm* Tg^C3001L zygosity. As shown in the bottom of Fig 4B, the Tg/WT ratio for mouse C7 is 1.32. By comparing the Tg/WT ratio of mice m12, m13, m14, and m15, their genotypes can readily be identified as homozygous, null, heterozygous, and homozygous for *Atm*Tg^C3001L transgene, respectively. Indeed, following-up with progeny testing confirmed the homozygous genotype determined by our allele-

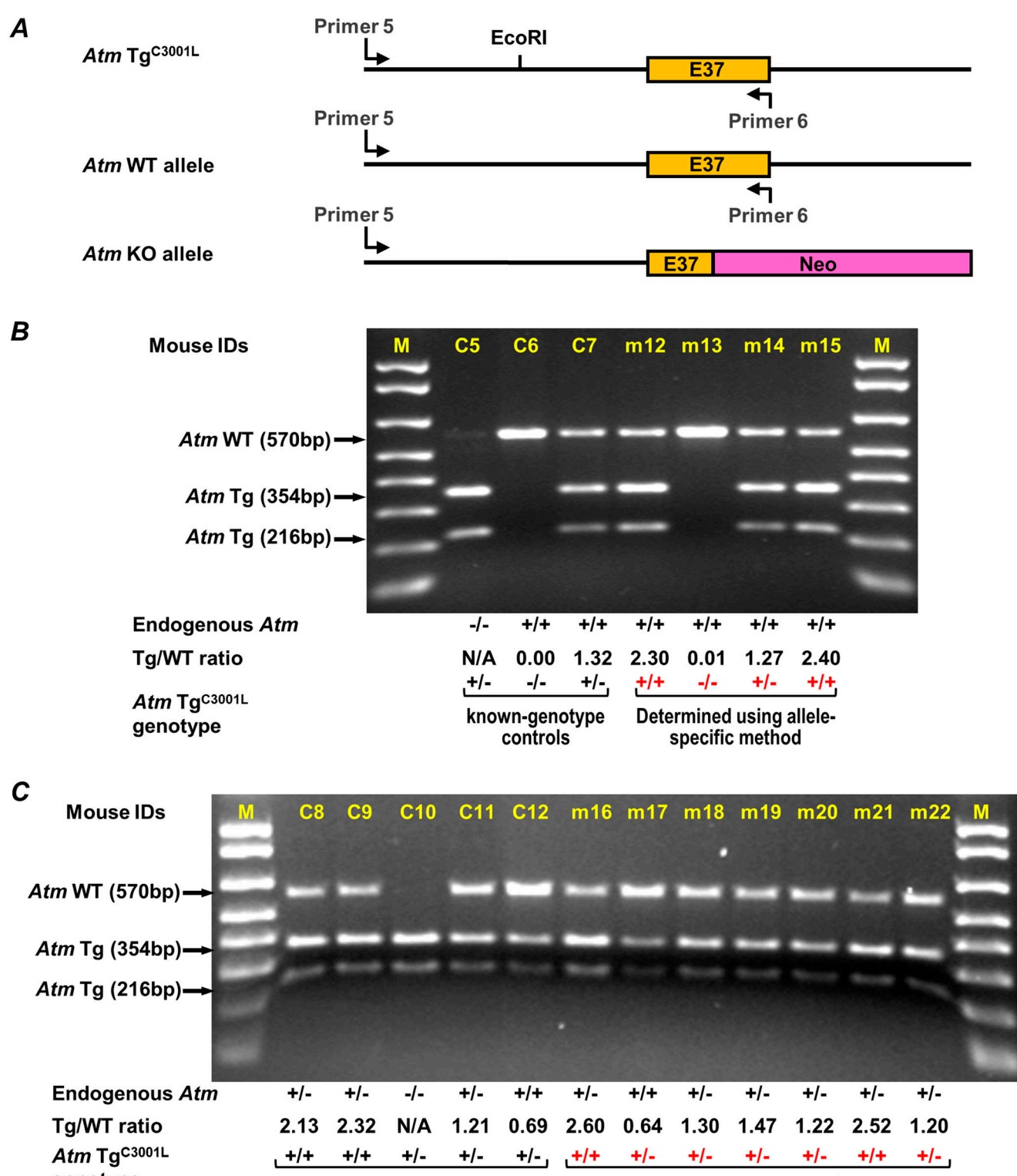

**Figure 4.  Genotyping *Atm* Tg^C3001L transgene zygosity using an allele-specific RE/PCR method.**
**(A)** Schematic illustration of *Atm* Tg^C3001L transgene, *Atm* WT allele, and *Atm* KO allele. Their sequences are aligned in reference to exon 37 (E37). Primers 5 and 6 are used to perform PCR, amplifying a DNA segment in both *Atm* Tg^C3001L transgene and *Atm* WT allele, but not in *Atm* KO allele. **(B)** The working principle of the allele-specific genotyping method. Mice with known genotype of transgene were used as controls (C5 to C7). DNA fragments from EcoRI digestion were analyzed in a 1.4% agarose gel. Below the gel are listed the genotype of *Atm* WT allele, the ratio of band intensities between Tg and WT alleles (Tg/WT ratio), and the genotype of *Atm* Tg^C3001L transgene for control (black) and assayed mice (assay-determined genotype is in red color, m12–m15). **(C)** A representative result for genotyping seven mice, m16 to m22, using the allele-specific method. M, 1 Kb plus DNA ladder (only <1,000-bp bands are shown) (Invitrogen).

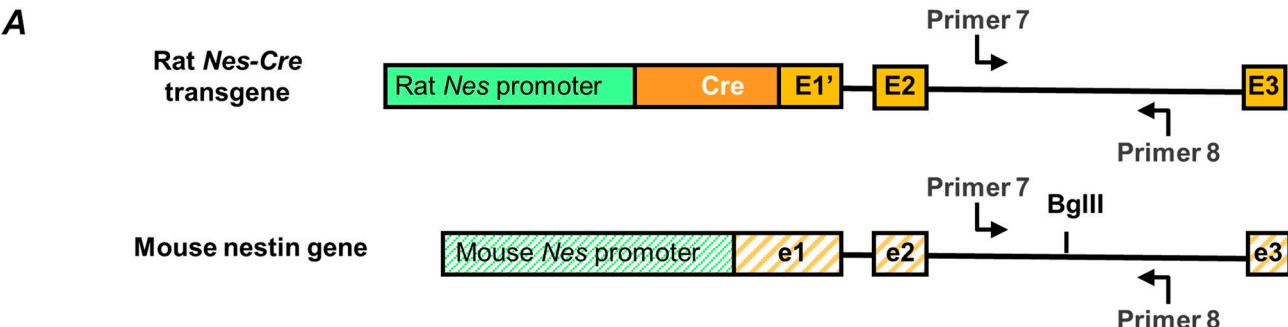

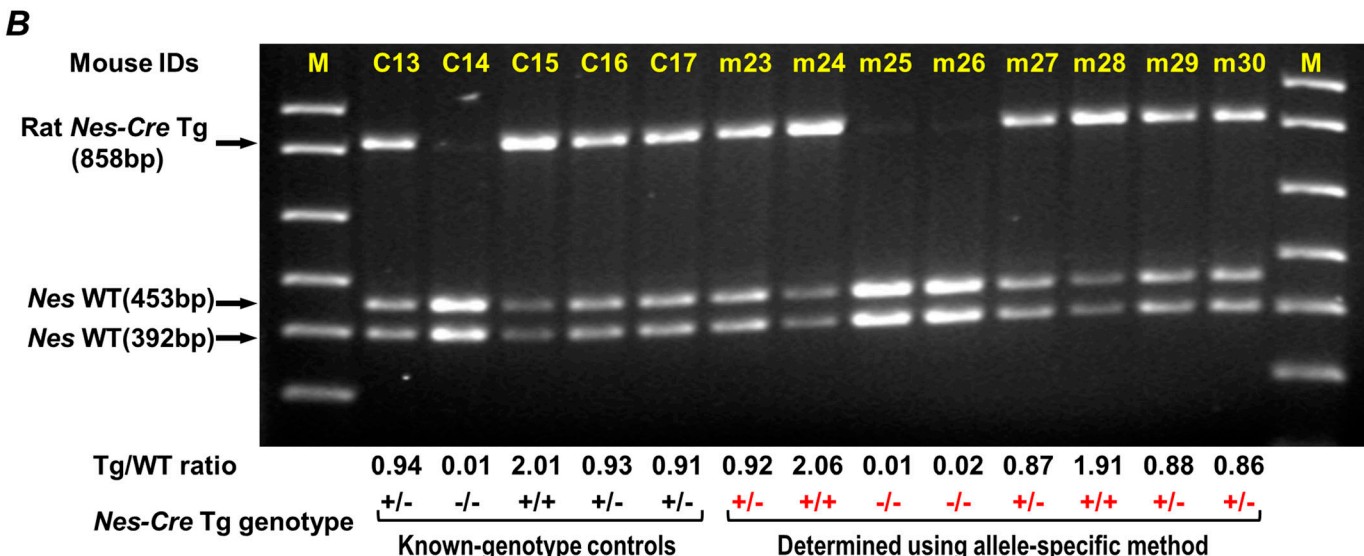

**Figure 5.  Genotyping *Nes-Cre* transgene zygosity using the allele-specific RE/PCR method.**
**(A)** Schematic illustration of a rat *Nes-Cre* transgene and the mouse *Nes* gene. Their sequences are aligned in reference to exon 2 (E2 for rat and e2 for mouse). Primers 7 and 8 are used to amplify a DNA segment in rat and mouse *Nes* intron 2. **(B)** A representative result for genotyping eight mice, m23 to m30, using the allele-specific genotyping method. Mice with known genotype of *Nes-Cre* transgene were used as controls (C13 to C17). DNA fragments from BglII digestion were analyzed in a 1.4% agarose gel. Below the gel are listed the ratio of band intensities between Tg and WT alleles (Tg/WT ratio) and the genotype of *Nes-Cre* transgene for control mice (black) and for mice m23–m30 (red). M, 1 Kb plus DNA ladder (only <1,000-bp bands are shown) (Invitrogen).

product generated from the mouse genome (Fig 5A). To prevent a situation where the majority of the PCR products are generated from the *Nes-Cre* transgene due to its high tandem repeat, we first determined its repeat number by performing RE/PCR using sequentially diluted *Nes-Cre* heterozygous DNA. We found that both rat and mouse templates were efficiently amplified at a 1:300 dilution (Fig S1B). Thus, all DNA samples were diluted with C56BL/6 DNA by 1:300 before their use in allele-specific genotyping. Eight mice (m23 to m30), offspring of a crossing of *Nes-Cre* heterozygous mice, were then genotyped using the allele-specific method (Fig 5B). Known genotype control C14 showed a complete digestion of BglII to the PCR product generated from the *Nes-Cre* null mouse. Other four known genotype controls (C13, C15, C16, and C17) showed a nearly a twofold increase in Tg/WT ratio from heterozygous mice (C13, C16, and C17) to homozygous mouse (C15). Two of the eight genotype-unknown mice, m24 and m28, were identified as *Nes-Cre* homozygotes. Follow-up progeny testing confirmed the identity of m24 and m28 as *Nes-Cre* homozygotes (Fig S2B and C).

Similarly, for *Syn1-Cre* mice, we identified primers 9 and 10, located on the rat and mouse synapsin 1 gene promoter sequence,

and a unique restriction enzyme site SphI on the mouse template (Fig 6A). Assessment of *Syn1-Cre* transgene tandem repeat number showed that both rat and mouse templates were efficiently amplified at 1:100 dilution factor (Fig S1C). Fig 6B shows that the allele-specific genotyping method readily genotypes m31 to m35, offspring derived from crossing of *Syn1-Cre* heterozygous mice.

## Discussion

As transgenic mouse models continue to be an indispensable tool for virtually every aspect of biological research, simple and reliable methods for determining transgene zygosity are highly desirable. Current methods to determine transgene zygosity are either time consuming (such as progeny testing), technically challenging (such as NGS, computation biology-based analyses) or have considerable inaccuracy (such as the qPCR-based method). Here, we report a new allele-specific quantification method that is used to determine zygosities in mice. This method is rapid (<2 d) and highly accurate

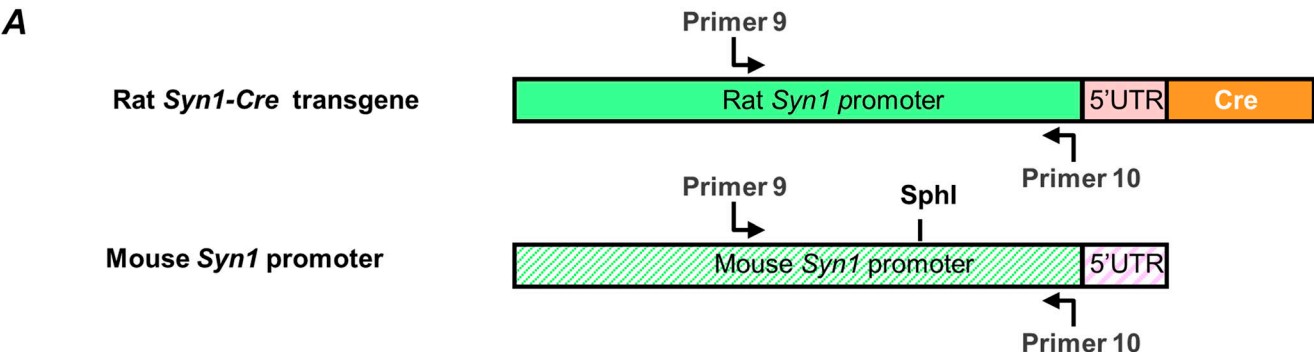

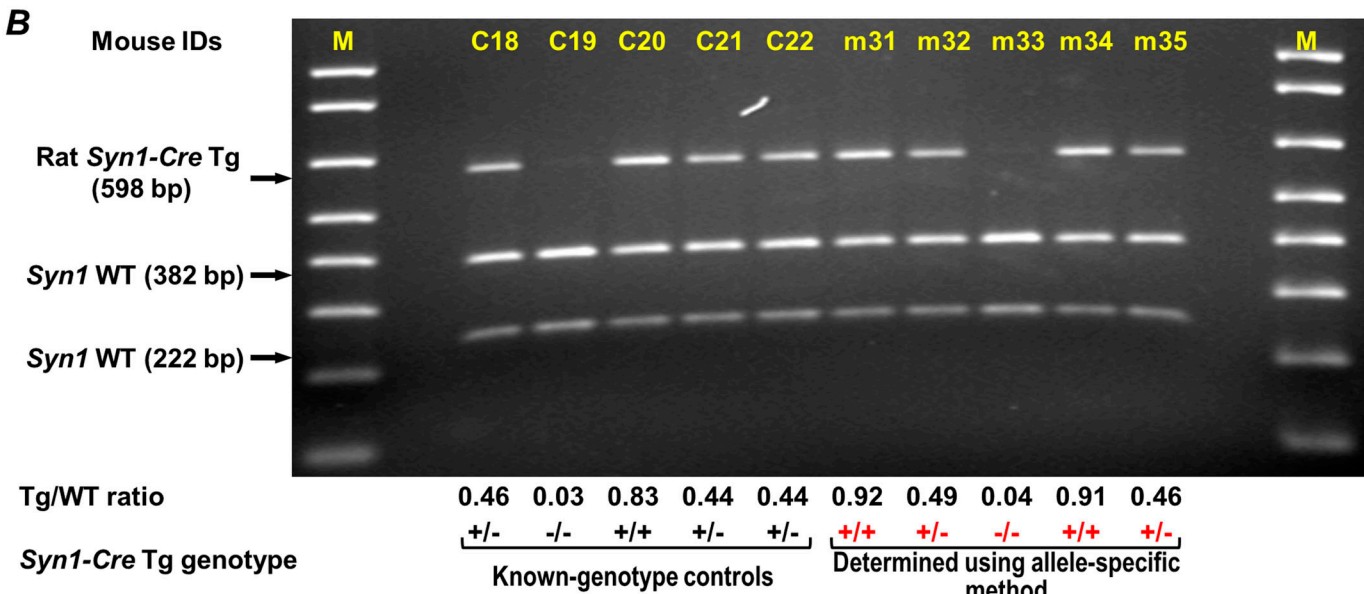

**Figure 6. Genotyping *Syn1-Cre* transgene zygosity using the allele-specific method.**
**(A)** Schematic illustration of a rat *Syn1-Cre* transgene and the mouse *Syn1* gene promoter. Their sequences are aligned in reference to 5′ UTR (5′ untranslated region). Primers 9 and 10 are used to amplify a DNA segment in rat and mouse *Syn1* promoter. **(B)** A representative result for genotyping five mice, m31 to m35, using the allele-specific genotyping method. Mice with known genotype of *Syn1-Cre* transgene were used as controls (C18 to C22). DNA fragments from SphI digestion were analyzed in a 1.4% agarose gel. Below the gel are listed the ratio of band intensities between Tg and WT alleles (Tg/WT ratio) and the genotype of *Syn1-Cre* transgene for control mice (black) and for mice m31–m35 (red). M, 1 Kb plus DNA ladder (only <1,000-bp bands are shown) (Invitrogen).

(>96%). It takes advantage of unique restriction enzyme sites within transgenes or their homologous sequences in the mouse genome to carry out allele specific restriction enzyme digestion of PCR products (RE/PCR). This is the first use of RE/PCR to genotype transgene zygosity where the insertion loci are unknown. In comparison, RE/PCR procedures have only been used in mice to determine the flanking sequence of a transgene (Bryda & Bauer 2010) or to genotype mutants with a known flanking sequence of the insertion loci (such as knock in) (Bruins et al, 2004; Willis et al, 2011). None of those procedures can be used to determine the zygosity of transgenes whose insertion loci are unknown. Using the *Atm* Tg^C3001L transgenic mouse as an experimental system, we developed a RE/PCR method to discriminate PCR products rising from *Atm* Tg^C3001L transgene versus endogenous WT *Atm* gene. The WT *Atm* gene is then used as an internal reference to quantify zygosity of the *Atm* Tg^C3001L transgene (Fig 7). We further showed that this

method can easily be applied to other transgenic models, such as *Nes-Cre* mice (Strain 016261; Jackson laboratory) and *Syn1-Cre* mice (Strain 003966; Jackson laboratory). Our results demonstrate that the DNA templates from transgenes and their homologs can be efficiently amplified simultaneously (single-tube PCR) even though these template sequences are not identical, and the allele-specific genotyping method can be used on any transgenic strains as long as these mice harbor a homologous sequence to the transgene and carry restriction enzyme sites unique to either the transgene or its homologous counterpart.

We surveyed the top 100 most frequently requested transgenic strains at Jackson Laboratory and identified 53 of these strains that are homozygous viable, for which our new transgene zygosity assay could potentially be applied. 79% of these 53 strains (42 out of 53) were created using transgenes containing DNA sequences originating from species other than mouse and

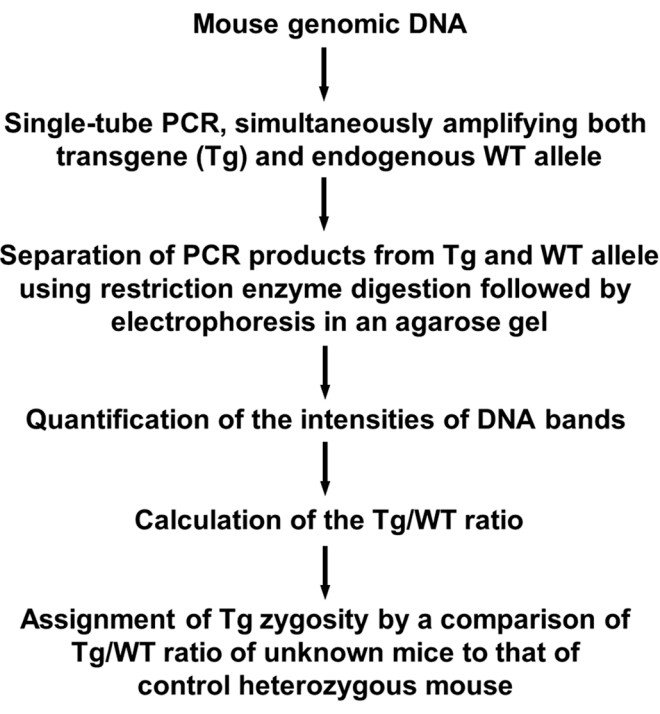

**Figure 7.  A flow chart of an allele-specific genotyping method for transgene zygosity determination in mice.**

homologous to their mouse counterparts. These strains, therefore, meet the requirements of our allele-specific genotyping method. As an example, a detailed summary highlights the sequences applicable to the allele-specific genotyping method in the first nine strains of the top 10 homozygous viable mice (Table 3). In the last strain (005023), where transgene sequence is from mouse and RE/PCR method is not applicable, other methods such as progeny testing, computation biology-based analyses, and qPCR are still valuable alternatives for genotyping transgene zygosity.

Importantly, the allele-specific genotyping method is as accurate as progeny testing, the gold standard for determining transgene zygosity, but takes 2 d instead of a month or more. This method is also readily scalable and suitable for processing large numbers of specimens rapidly, something that cannot be done with progeny testing. Moreover, compared with the most widely used method of transgene zygosity determination, qPCR analysis, allele-specific genotyping is much more accurate, as we demonstrated here (Table 2). This is because of three features of the allele-specific method described here. First, the internal reference gene has a DNA sequence similar to the transgene. Second, both templates are amplified under the same PCR condition with near identical amplification efficiency. Third, the allele-specific quantification analyses reaction products at the plateau phase of PCR reaction, and thus is insensitive to the unavoidable variations in template quantity among specimens. Therefore, the yield of PCR products from the transgene and reference gene reliably reflects their template quantity. In contrast, qPCR reactions for transgene and the reference gene are performed in two separate reactions, which inevitably introduces variations. Also, qPCR measures reaction products in the

exponential phase of PCR reaction and is consequently highly sensitive to variations in template quantity among specimens. Therefore, allele-specific quantification method is not only fast and simple, but also has high accuracy when used to determine transgene zygosity in mice. Finally, we believe the allele-specific quantification method we developed here represents a superior alternative to the less accurate qPCR and time-consuming progeny testing for transgene zygosity. It also does not depend upon genome sequencing and computational biology methods.

Allele-specific quantification is a versatile method. As shown here, the *Atm* Tg$^{C3001L}$ transgene can also be used as an internal reference to determine the zygosity of the endogenous *Atm* gene. Given that the *Atm* Tg$^{C3001L}$ transgene is bred on to WT mice, typical PCR protocols used to genotype zygosity of WT *Atm* gene will be compromised by the presence of the *Atm* Tg$^{C3001L}$ transgene. The allele-specific quantification method can be used to discriminate mice with a genotype of *Atm*$^{+/+}$, *Atm* Tg$^{C3001L+/-}$ from those of *Atm*$^{+/-}$, *Atm* Tg$^{C3001L+/-}$ (Fig 4C, m17 versus m18). With consideration of parent mouse genotypes, additional combinations of WT *Atm* allele, *Atm* KO allele, and *Atm* Tg$^{C3001L}$ transgene can be readily determined as well. For a situation where *Atm* Tg$^{C3001L}$ transgene is bred onto an *Atm*$^{-/-}$ background, a new primer (primer 6b) can be used to perform allele-specific zygosity determination (Fig S3A and B). This method is cost-effective and time-saving when used to investigate the biological role of a mutated *Atm* transgene in the endogenous *Atm* null background.

## Materials and Methods

Animals- *Atm* Tg$^{C3001L}$ transgenic mice were generated via pronuclear injection of a BAC that contained the mouse ATM gene with a C→L mutation corresponding to amino acid 3001 position. This transgenic strain, designated as *Atm* Tg$^{C3001L}$, was created using fertilized C57BL/6 × 129 oocytes. Positive founder mouse was backcrossed with C57BL/6 mice for five generations to segregate possible multiple insertion events (Fig S4). Resultant *Atm* Tg$^{C3001L}$ mice were maintained on a mixed genetic background of C57BL/6 × 129. *Atm* Tg$^{C3001L}$ carriers were identified using PCR with primer 1 (5′-AGCACAACCACACTGAATGC-3′) and primer 2 (5′-GTTTTTTGCGATCTG CCGTTTC-3′) (Fig 1A). *Nes-Cre* mice (Strain 016261; Jackson laboratory) and *Syn1-Cre* mice (Strain 003966; Jackson laboratory) were purchased from Jackson laboratory. *Nes-Cre* transgene was identified using PCR with primer 11 (5′-ATGCAACGAGTGATGAGG-3′) and primer 12 (5′-ATCAACGTTTTCTTTTCGGATC-3′). Homozygous transgenic mice were generated via breeding of heterozygous mice. All animal experiments were performed in agreement with the Guide for the Use and Care of Laboratory Animals.

Extraction and preparation of mouse genomic DNA—mouse genomic DNA was extracted from tail biopsies following the protocol detailed in the PrepEase Genomic DNA Isolation Kit (USB). Briefly, mouse tails were digested with proteinase K, and genomic DNA was extracted and precipitated. The resultant DNA pellet was rehydrated overnight in 80-μl water supplemented with 10 μg/ml RNase A (USB) with gentle shaking. Genomic DNA was then quantified using a NanoDrop 1000 Spectrophotometer (Thermo Fisher Scientific), followed by dilution of the DNA with water to a working concentration of

**Table 3.  Transgene structure of the top 10 requested homozygous viable transgenic mice at Jackson laboratory.**

| Strain ID | Transgene symbol | Transgene structure |
|---|---|---|
| 003831 | Tg(TcraTcrb)1100Mjb | H2K$^b$ promoter_cDNA encoding the complete 149.42 $\alpha$-chain_**human β-globin gene** (part of exon 2/intron/exon 3/polyadenylation signal) |
| 014565 | Tg(FCGRT)32Dcr | A cosmid clone containing the complete **human FCGRT gene** |
| 030890 | Tg(IL15)1Sz | A human BAC (RP11-620F3) containing the complete **human interleukin 15 gene**. |
| 013062 | Tg(CMV-IL3,CSF2,KITLG)1Eav | Three separate transgenes each carrying either **the human interleukin-3 gene, the human granulocyte/macrophage-stimulating factor gene or the human steel factor gene**. |
| 004919 | Tg(CAG-FCGRT)276Dcr | The CMV enhancer, **chicken β-actin promoter, and intron 1, a cDNA sequence encoding the human FcRn α-chain, rabbit beta-globin intron**, and SV40 polyA sequence. |
| 003475 | Tg(HLA-A2.1)1Enge | A 7-kb EcoRI fragment containing the **full length human HLA-A2.1 gene**. |
| 006567 | Tg(CAG-EGFP)131Osb | The **chicken β-actin promoter** and CMV enhancer, β-actin intron, eGFP, and **bovine globin poly-adenylation signal**. |
| 004353 | Tg(UBC-GFP)30Scha | The **human ubiqutin C promoter**_eGFP |
| 006054 | Tg(CMV-cre)1Cgn | The human CMV promoter_Cre coding sequence_**rabbit β-globin gene intron**/poly A signal |
| 005023 | Tg(TcraTcrb)8Rest | A cosmid clone (cos HYβ9-1.14-5) containing rearranged mouse α-chain and β-chain of the TCR |

Transgenes of the first nine strains contain DNA sequences originated from species other than mouse. Those sequences are homologous to their mouse counterparts and are highlighted in bold in the table. Transgene sequence of the last strain (005023) was from mouse DNA.

~20 ng/μl. The precise concentration of this working solution was redetermined using the NanoDrop 1000 Spectrophotometer before using it as the template for qPCR. When assessing the tandem repeat number of transgenes, DNAs from heterozygous mice and C57BL/6 mice were quantified and then diluted with water to a final concentration of 30 ng/μl. Then, the heterozygous mouse DNA solution (30 ng/μl) was mixed with C57BL/6 mouse DNA solution (30 ng/μl) at various ratios, ranging from 1:1 up to 1:1,000 depending on the number of tandem repeats of the transgene present at the insertion loci. RE/PCR was performed using the mixed DNA to identify a dilution condition where both the transgene and its endogenous homolog were efficiently amplified (Fig S1).

Primer design for *Atm* Tg$^{C3001L}$ qPCR—primers for qPCR were designed following the protocol outlined by Bustin and Huggett (2017). In brief, the target sequence was selected on the BAC construct corresponding to the junction between the mouse sequence and vector sequence (Fig 1A). Five sets of primers were identified 5′ and 3′ to this junction sequence. Using a computer program OligoAnalyzer (IDT), primers lacking stable structures including hairpins, self-dimers, and heterodimers were selected, and subjected to a second round of screening using the NCBI BLASTn suite to identify those sharing a low similarity (<40%) to off-target sequence(s) in the mouse genome. qPCR for *Atm* Tg$^{C3001L}$ transgene was performed using primer 3 (5′-AAT-GATTATCTCAGGCACAAATATCACAGGTCTTCT-3′) and primer 4 (5′-GAATT-GACTAGTGGGTAGGCCTGGCG-3′). Primer 4 is located on the BAC vector sequence that shares low similarity to the mouse genome (Fig 1A), ensuring high specificity of the qPCR reaction. qPCR primers for 18S rRNA are 5′-CAAAGATTAAGCCATGCATGTCTAAGTACGC-3′ and 5′-GGCATGT ATTAGCTCTAGAATTACCACAGTTATCC-3′.

Determination of *Atm* Tg$^{C3001L}$ zygosity using qPCR–qPCR reactions were performed using 96-well plates on a CFX96 Touch Real-Time PCR Detection System (Bio-Rad). In brief, the reaction was carried out in a 7-μl mixture containing 3.5 μl iQ SYBR Green Supermix solution (Bio-Rad), 15 ng genomic DNA, 270 nM forward primer, 270 nM reverse primer, 0.3 μl 10X enhancer solution (Invitrogen). The reaction solution was sealed with one drop of light mineral oil (Thermo Fisher Scientific) during PCR. The qPCR program had one cycle of heating (95°C for 2 min), then 40 cycles of 94°C for 6 s and 60°C for 10 s, followed by one cycle of melting curve measurement. All DNA samples were measured in triplicate. The reference gene was 18S rRNA. At the end of qPCR, *Atm* Tg$^{C3001L}$ zygosity was determined using the following equations according to the method outlined by Tesson et al (2010).

Cycle threshold (Ct) value
= average of the triplicate Ct values for each sample

$$\Delta Ct = Ct_{Atm\ transgene} - Ct_{18S\ rRNA}$$

$$\Delta\Delta Ct = \left(Ct_{Atm\ transgene} - Ct_{18S\ rRNA}\right)_{unknown\ mouse} - \left(Ct_{Atm\ transgene} - Ct_{18S\ rRNA}\right)_{Atm\ Tg\ Het\ mouse}$$

*Atm* Tg$^{C3001L}$ zygosity status
= 2$^{-\Delta\Delta Ct}$ (with values of 0.8-1.3 for het, and 1.8-2.3 for homo)

Determination of transgene zygosity using an allele-specific genotyping method—Transgene and its WT counterpart were simultaneously amplified in a single PCR reaction using the following primers.

*Atm* Tg$^{C3001L}$ (Figs 4A and S3A):

primer 5 (5′-GCAGATCCTAAGTAGGTGAGCT-3′)

primer 6 (5′-CGAATTTGCAGGAGTTGCTGAG-3′)

Primer 6b (5′-ACATCATGGATCAAGTATGGCAGC-3′)

*Nes-Cre* transgene (Fig 5A):

primer 7 (5′- AGGCAGGCAATCTCCAGTGTCTATG-3′)

primer 8 (5′- CAGGGGAAGTGGGAATTCTCAGG-3′)

*Syn1-Cre* transgene (Fig 6A):

primer 9 (5′- CGCCTGTCTGGTGATGTTTACGC-3′)

primer 10 (5′- GCCGCAGAGCGTATGGTCG-3′)

Resultant PCR products were purified using the QIAquick PCR purification kit (QIAGEN), followed by a digestion of ~90 ng purified PCR products with the restriction enzyme EcoRI (NEB) for *Atm* Tg$^{C3001L}$ mice, BglII (NEB) for *Nes-Cre* mice, and SphI (NEB) for *Syn1-Cre* mice at 37°C for 8–12 h. Restriction enzyme-cut DNA fragments were then subjected to electrophoresis using agarose gels. The intensity of each DNA band was quantified using image J analysis software. The ratio of transgene to WT allele (Tg/WT ratio) was calculated using the following equation:

$$\text{Tg/WT ratio} = \text{sum of Tg band intensity/sum of WT band intensity}$$

(Figs 4B and C, 5B, 6B, and S3B)

To determine the transgene zygosity status of mice, the Tg/WT ratio of each mouse is compared with the value observed of heterozygous mice included as a control in each reaction. If mice have a Tg/WT value near (within 25%) that of the heterozygous control, the mice are heterozygotes. If their Tg/WT value is double this value, the mice are homozygotes.

## Data Availability

The authors confirm that the data supporting the findings of this study are available within the article.

## Supplementary Information

## Acknowledgements

This work was supported by NIH CA161882 (RA Fisher), NIH AA025919 (RA Fisher). We would like to thank Yufang Kong and Dr Stefan Strack for sharing *Syn1-Cre* mice.

### Author Contributions

J Yang: conceptualization, data curation, formal analysis, investigation, methodology, and writing—original draft, review, and editing.

AN DeVore: conceptualization, data curation, formal analysis, validation, investigation, and writing—original draft, review, and editing.

DA Fu: conceptualization, data curation, formal analysis, validation, investigation, and writing—review and editing.

MM Spicer: conceptualization, formal analysis, and validation.

M Guo: formal analysis, validation, and writing—review and editing.

SG Thompson: formal analysis, validation, and writing—review and editing.

KE Ahlers-Dannen: conceptualization, investigation, and methodology.

F Polato: conceptualization, data curation, formal analysis, and investigation.

A Nussenzweig: conceptualization, data curation, formal analysis, and investigation.

RA Fisher: conceptualization, resources, data curation, formal analysis, supervision, funding acquisition, validation, investigation, methodology, project administration, and writing—original draft, review, and editing.

### Conflict of Interest Statement

The authors declare that they have no conflict of interest.

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
