## [Reviewer comments · Life Science Alliance]

Rapid and precise genotyping of transgene zygosity in mice using an allele specific method

Jianqi Yang, Alison DeVore, Daniel Fu, Mackenzie Spicer, Mengcheng Guo, Samantha Thompson, Katelin Ahlers-Dannen, Federica Polato, Andre Nussenzweig and Rory Fisher

DOI: 10.26508/lsa.202201729

Corresponding author(s): Dr. Rory Fisher (Univ of Iowa)

Review timeline:

Submission Date:	2022-09-19
Editorial Decision:	2022-11-07
Revision Received:	2023-02-06
Editorial Decision:	2023-03-01
Revision Received:	2023-03-20
Editorial Decision:	2023-03-21
Revision Received:	2023-03-22
Accepted:	2023-03-23

Scientific Editor: Novella Guidi

Transaction Report:

No Peer Review Process File is available with this article, as the authors have chosen not to make the review process public in this case.

Re: Life Science Alliance manuscript #LSA-2022-01729-T

Rory Fisher
Univ of Iowa

Dear Dr. Fisher,

Thank you for submitting your manuscript entitled "Rapid and precise determination of transgene zygosity in mice using an allele-specific genotyping method" to Life Science Alliance. The manuscript was assessed by expert reviewers, whose comments are appended to this letter. We invite you to submit a revised manuscript addressing all the Reviewer comments.

Thank you for this interesting contribution to Life Science Alliance. We are looking forward to receiving your revised manuscript.

Sincerely,

B. MANUSCRIPT ORGANIZATION AND FORMATTING:

Re: Life Science Alliance manuscript #LSA-2022-01729-TR

Dr. Rory Fisher
Univ of Iowa
Department of Pharmacology
2-512 Bowen Science Building
Iowa City, IA 52242

Dear Dr. Fisher,

Thank you for submitting your revised manuscript entitled "Rapid and precise genotyping of transgene zygosity in mice using an allele-specific method" to Life Science Alliance. The manuscript has been seen by the original reviewers whose comments are appended below. While the reviewers continue to be overall positive about the work in terms of its suitability for Life Science Alliance, some important issues remain.

Our general policy is that papers are considered through only one revision cycle; however, given that the suggested changes are relatively minor, we are open to one additional short round of revision. Please note that I will expect to make a final decision without additional reviewer input upon resubmission.

Please submit the final revision within one month, along with a letter that includes a point by point response to the remaining reviewer comments.

B. MANUSCRIPT ORGANIZATION AND FORMATTING:

Sincerely,

3rd Editorial Decision

21 March 2023

RE: Life Science Alliance Manuscript #LSA-2022-01729-TRR

Dr. Rory Fisher
Univ of Iowa
Department of Pharmacology
2-512 Bowen Science Building
Iowa City, IA 52242

Dear Dr. Fisher,

Thank you for submitting your revised manuscript entitled "Rapid and precise genotyping of transgene zygosity in mice using an allele-specific method". We would be happy to publish your paper in Life Science Alliance pending final revisions necessary to meet our formatting guidelines.

-please add the Twitter handle of your host institute/organization as well as your own or/and one of the authors in our system

A. FINAL FILES:

-- Summary blurb (enter in submission system): A short text summarizing in a single sentence the study (max. 200 characters including spaces). This text is used in conjunction with the titles of papers, hence should be informative and complementary to the title. It should describe the context and significance of the findings for a general

readership; it should be written in the present tense and refer to the work in the third person. Author names should not be mentioned.

B. MANUSCRIPT ORGANIZATION AND FORMATTING:

Thank you for your attention to these final processing requirements. Please revise and format the manuscript and upload materials within 3 days.

Sincerely,

RE: Life Science Alliance Manuscript #LSA-2022-01729-TRRR

Dr. Rory Fisher
Univ of Iowa
Department of Pharmacology
2-512 Bowen Science Building
Iowa City, IA 52242

Dear Dr. Fisher,

Thank you for submitting your Methods entitled "Rapid and precise genotyping of transgene zygosity in mice using an allele-specific method". It is a pleasure to let you know that your manuscript is now accepted for publication in Life Science Alliance. Congratulations on this interesting work.

DISTRIBUTION OF MATERIALS:

Again, congratulations on a very nice paper. I hope you found the review process to be constructive and are pleased with how the manuscript was handled editorially. We look forward to future exciting submissions from your lab.

Sincerely,
